# Identification and Antimicrobial Resistance in *Klebsiella* spp. Isolates from Turkeys in Poland between 2019 and 2022

**DOI:** 10.3390/ani12223157

**Published:** 2022-11-15

**Authors:** Joanna Kowalczyk, Ilona Czokajło, Marta Gańko, Marcin Śmiałek, Andrzej Koncicki

**Affiliations:** 1Department of Poultry Diseases, Faculty of Veterinary Medicine, University of Warmia and Mazury, Oczapowskiego 13, 10-719 Olsztyn, Poland; 2SLW Biolab Veterinary Laboratory, ul. Grunwaldzka 62, 14-100 Ostróda, Poland

**Keywords:** *Klebsiella* spp., antimicrobial resistance, turkeys

## Abstract

**Simple Summary:**

The increasing incidence of multidrug-resistant microbes is a major problem in human and veterinary medicine. In our study we assessed the prevalence and antibiotic resistance of *Klebsiella* spp. strains isolated from turkey broilers and breeders. The obtained results show that isolates have become more susceptible to most of the popular antimicrobials.

**Abstract:**

One of the important problems in poultry production is bacterial infections and increasing resistance to antibiotics. The increasing incidence of multidrug-resistant bacteria is a major challenge for physicians and veterinarians and considerably limits treatment options. This study was undertaken in order to assess the prevalence and antibiotic resistance of *Klebsiella* spp. strains isolated from turkeys sampled from 2019 to 2022 in Poland. The material for study consisted of clinical samples taken during routine monitoring and microbiological identification testing at commercial poultry farms. From all 507 isolates of *Klebsiella*, 95% were identified by MALDI-TOF (Matrix-Assisted Laser Desorption - Ionisation-Time of Flight) as *Klebsiella pneumonia*, 2% were *Klebsiella oxytoca*, 2% *Klebsiella variicola*, or unidentified (1%). All isolated *Klebsiella* strains were tested for antimicrobial susceptibility by disk diffusion. The results of our study indicated that colistin, neomycin, florfenicol and amoxicillin/clavulanic acid were the most effective against the *Klebsiella* spp. isolated from turkeys. In addition, the results show a decrease in the number of multi-resistant *Klebsiella* spp. strains between 2019 and 2021.

## 1. Introduction

The spread of antibiotic resistant bacteria is a global challenge to both human and veterinary medicine. According to the Eleventh European Surveillance of Veterinary Antimicrobial Consumption Report titled “Sales of veterinary antimicrobial agents (VMP) in 31 European countries in 2019 and 2020” [1], the sales of antimicrobial veterinary medicinal products (VMPs) for food-producing animals reached 5577.8 tons in Europe (including 856.7 tons in Poland alone), the bulk of which comprised penicillins and tetracyclines. Although there has been a global decline in sales of antimicrobial drugs over the years, consumption in Poland has only changed for specific drug groups (increase in sales of penicillins and tetracyclines, decrease in lincosamides and sulfonamides). Unfortunately, the irresponsible use of drugs, especially in empirical therapies and popular metaphylactic treatments, has led to the proliferation of multidrug-resistant bacteria among humans and animals, which presents a serious epidemiological risk. Industrial poultry production is rife with bacterial infections resistant to commonly used chemotherapeutics, which severely complicates treatment. The bacteria most commonly isolated from such infections are those of the order Enterobacterales and genus Enterobacteriaceae, most commonly *E. coli* and *Salmonella* spp. These have been identified as major pathogens with established roles in poultry disease [2]. There are also concerning data pointing to a rise in antibiotic-resistant isolates of opportunistic Enterobacteriaceae including *Klebsiella* spp., *Proteus* spp. and *Enterobacter aerogenes*. These species naturally live in the digestive mucosa but have also been isolated from excretory system infections, respiratory infections, and other infections [3].

*Klebsiella* spp. are Gram-negative, non-sporulating, non-ciliated bacilliform bacteria with a thick cell wall, which is responsible for the bacterium’s high virulence in vivo and the mucoid appearance of its colonies in solid-state in vitro cultures. Multiple *Klebsiella* species and sub-species have been identified, *Klebsiella pneumoniae* being considered the most clinically significant in both humans and animals. *Klebsiella* spp. are classified as contaminants of surface water, plants, soil, wastewater, and other environments. This, combined with their drug resistance, creates a risk of resistance genes being transferred to other microorganisms (especially genes conferring resistance to carbapenems), as well as a risk of their potentially causing superinfections and exacerbating primary infections in immunocompromised individuals [4,5]. Furthermore, *Klebsiella* spp. in poultry, and ultimately in retail poultry food products, can pose a serious risk to consumer health. *Klebsiella* spp. can cause severe pneumonia, urinary tract infections, endocarditis, liver abscesses, and even septicemia in humans [6,7]. In birds, *Klebsiella* spp. have been isolated from dead embryos, omphalitis, yolk sac infections, dermatitis, cellulitis, inflamed respiratory mucosa, and inflamed ascites. The standard operating procedure in Poland is to treat such infections with chemotherapeutics such as colistin and trimethoprim-potentiated sulfonamides as the first line of treatment and enrofloxacin as the second [8]. Unfortunately, *Klebsiella* spp. isolates often exhibit multiple resistance to chemotherapeutics. Additionally, the presence of genes coding for extended-spectrum β-lactamase (ESBL) in *Klebsiella* spp. renders ineffective the β-lactam antibiotics widely used to treat bacterial infections in poultry.

A review of the available studies on *Klebsiella* spp. resistance indicates that the problem is a global one. This conclusion is supported by the fact that the last 10 years have brought a noted increase in research on this subject, the output within this timeframe totaling approx. 10 thousand papers [9]. As a country where *Klebsiella* spp. infections in poultry must often be treated, Poland is not spared the problem of the resistance of some strains of the bacteria to antimicrobials. *Klebsiella* spp. are often isolated from birds suffering from polyetiological diseases. This bacterium is also often isolated from surface swabs in poultry hatcheries and slaughterhouses, which stimulates further interest in resistant strains and potential modes of treatment that would prevent further antibiotic resistance without sacrificing efficacy.

With this in mind, a study was conducted to assess the prevalence and antibiotic resistance of *Klebsiella* spp. strains isolated from turkey broilers and breeders sampled from 2019 to 2022 in Poland, primarily in its north-eastern region.

## 2. Materials and Methods

### 2.1. Sample Collection and Bacterial Isolation

The material tested for *Klebsiella* spp. consisted of clinical samples taken between 2019 and 2022 during routine monitoring and microbiological identification testing at commercial poultry farms (most of which were located in north-eastern Poland). The subject flocks comprised birds from 1 day old to 44 weeks old. Samples were taken from live birds as palatine fissure swabs or trachea swabs (n = 153), as well as post-mortem from the lungs, heart, liver, spleen, suborbital sinuses, yolk sac, and joints (n = 354).

The samples were plated on MacConkey agar (Lab-Agar, Biomaxima, Lublin, Poland) and incubated under aerobic conditions at 37 °C ± 1 °C for 24 h ± 3 h. A manual inspection identified suspicious colonies (those which were pink and mucoid), which were subsequently analyzed with matrix-assisted laser desorption/ionization–time-of-flight mass spectrometry (MALDI-TOF) (Biotyper, Bruker, Billerica, MA, USA) and compared against reference spectra, which in most cases resulted in successful identification of *Klebsiella* species.

### 2.2. Antibiotic Sensitivity Testing and Detection of Extended-Spectrum β-Lactamases

All isolated and identified *Klebsiella* spp. strains were tested for antimicrobial susceptibility by disk diffusion. The assay was performed on Mueller–Hinton agar (Lab-Agar, Biomaxima, Lublin, Poland) with antibiotic disks from OXOID (Basingstoke, UK). Anti-microbial susceptibility was tested using 12 active substances: amoxicillin (AMX, antibiotic concentration 25 µg), amoxicillin/clavulanic acid (AMC, 30 µg), colistin (CL, 10 µg), doxycycline (DOX, 30 µg), enrofloxacin (ENR, 5 µg), florfenicol (FLO, 30 µg), flumequine (UB, 30 µg), lincomycin/spectinomycin (L/SPE, 109 µg), neomycin (NEO, 30 µg), oxytetracycline (OT, 30 µg), sulfamethoxazole/trimethoprim (SXT, 25 µg) and sulfonamides (SSS, 300 µg). The tested bacteria were suspended in sterile deionized water to a final turbidity of 0.5 McFarland. After the inoculum was plated on the agar, the diagnostic disks were applied, and the plates incubated for 16 to 18 h at 33–37 °C. *Klebsiella* spp. isolates were evaluated for their susceptibility to the tested panel of antimicrobials according to the Clinical and Laboratory Standards Institute guidelines [10]. As per the definition, strains were deemed multidrug-resistant if resistant to at least three antimicrobial groups.

The two-disk method was used to identify *Klebsiella* spp. capable of producing ESBLs, following a pre-established procedure [11]. Antibiotic disks were briefly applied to Mueller–Hinton agar with the bacterial suspension (Argenta, Poznań, Poland). The central disk contained amoxicillin with clavulanic acid; additional disks were placed containing ceftazidime and cefotaxime at both sides of the central disk (2 cm away from its center). The plates were incubated for 16–18 h at 33–37 °C. For an isolate to test positive, the zone of inhibition around the ceftazidime or cefotaxime disk had to be significantly larger near the central amoxicillin and clavulanic acid disk.

The following reference strains were used in the method of determining the drug susceptibility of bacteria of the order Enterobacterales: *Escherichia coli* ATCC 25922 (main quality control), *Escherichia coli* ATCC 35218 (beta-lactamase inhibitor control, ESBL negative) and *Klebsiella pneumoniae* subsp. *pneumoniae* ATCC 700603 (ESBL positive).

### 2.3. Statistical Analysis

A chi-squared test was used to determine significant differences between the numbers of field-sampled *Klebsiella* spp. strains susceptible to different antibiotics. All isolates for a given year obtained in 2019–2022 were included in the test. Statistical analysis was performed using Statistica 13.1 (StatSoft Polska, Kraków, Poland) with differences considered significant at *p* ≤ 0.05.

## 3. Results

A total of 507 *Klebsiella* spp. strains were isolated from the samples delivered to the laboratory between 2019 and 2022. A 95% of the strains were identified by MALDI-TOF as *Klebsiella* pneumonia and the remaining 5% were *Klebsiella oxytoca* (2%), *Klebsiella variicola* (2%), or unidentified (1%). The antibiotic sensitivity of the tested strains is given in Table 1, classifying them as sensitive, intermediate, or resistant. Table 2 shows the data broken down by year. 

Our data show that colistin (92.9% of strains susceptible), neomycin (90.14%), florfenicol (88.56%) and amoxicillin/clavulanic acid (82.6%) were the most effective against the *Klebsiella* spp. isolates. All of the *Klebsiella* spp. stains proved amoxicillin resistant. A small proportion of the strains (3.3%) were also able to produce ESBLs.

Table 3 indicates statistically significant differences in the numbers of *Klebsiella* spp. strains resistant to the tested active substances in the years 2019 to 2022. There were significant differences between the number of amoxicillin resistant *Klebsiella* spp. strains and the number of strains resistant to other active substances (*p* ≤ 0.05) in each analyzed year. No significant differences were found between the percentages of strains resistant to other antibiotics (with the exception of amoxicillin resistance, as noted above).

According to the drug-resistance data, the proportion of multidrug-resistant *Klebsiella* spp. in all isolates across all years was 38.2%. The percentage of multidrug resistant strains changed significantly throughout the years of study in being 35.9%, 43.48% and 35.88% in 2019, 2020 and 2021, respectively. Figure 1 shows trends in *Klebsiella* spp. resistance to different antimicrobials between 2019 and 2021. The percentage of amoxicillin-resistant strains was a constant 100% throughout the study. The percentage of strains resistant to most of the tested active substances increased from 2019 to 2020. This was followed by a drop in 2021 in strains resistant to most active substances, with the exceptions of colistin and neomycin.

## 4. Discussion

The rise in drug resistance is a global issue and a critical challenge for human and veterinary medicine [9]. In response to this threat, the WHO has launched the “One Health” initiative, which aims to combat the most pressing health issues facing the modern world, such as zoonoses, antibiotic resistance, and food safety. The initiative calls for measures aiming, inter alia, to limit antibiotic use through public awareness campaigns, to promote antibiotic-free therapies, to adopt phytogenics, and in particular to implement targeted therapies preceded by microbiological tests and antibiograms [12,13,14,15,16,17,18].

The increasing incidence of multidrug-resistant microbes is a major challenge for physicians and considerably limits treatment options. Certain microbes in this group, namely ESBL-producing Enterobacteriaceae, which include *Escherichia coli* and *Klebsiella* spp., are frequently isolated from healthy poultry and other animals. Furthermore, these bacteria are quick to proliferate in animal populations even in birds not subjected to any antibiotic treatment, which is all the more alarming [19,20,21]. Apart from vertical transmission, contamination by contact is also a route by which ESBL strains colonize new host material: comparative genome analyses of isolates from the environment and from carcass surfaces have shown that cross-contamination of meat can also occur directly in slaughterhouses. That is why poultry products, especially broiler chicken meat, have been investigated as a main source of multidrug-resistant bacteria dangerous to human health [15,22,23,24,25,26,27].

The available literature indicates that *Klebsiella* spp. are often found on poultry carcasses during and after slaughter [27,28]. In this case, the main source of contamination (including cross-contamination) is infection of birds during primary production and within the slaughterhouse itself. The finding that 3% of the turkey flocks analyzed in the present study were infected with ESBL-producing bacteria is very alarming, especially since we only considered *Klebsiella* spp. populations. This represents a major risk to public health in terms of infection from consuming undercooked food, and more importantly with regard to potential transfer of antibiotic resistance genes.

Our study shows that out of all the substances considered, colistin and neomycin are the most effective against *Klebsiella* spp. isolated from turkeys (Table 1). This is in line with the findings made by Majewski et al. [28], who additionally reported similar susceptibility to trimethoprim-potentiated sulphonamides as a characteristic of approx. 70% of *Klebsiella* spp. isolates from chickens in Poland. Our findings only diverge with respect to *Klebsiella* spp. susceptibility to florfenicol and amoxicillin/clavulanic acid in our own study, over 80% of the isolates exhibited antibiotic sensitivity (Table 1), whereas Majewski et al. [28] found only 10% and 38% of the strains to be sensitive to the two antimicrobials, respectively. It should be noted that the study by Majewski et al. [21] concerned *Klebsiella* spp. isolated from hens, which may point to differences in bacterial resistance between different species of poultry. We can also cite Wu et al. [27] as further confirmation of the species factor, as their study showed 96.7% of *Klebsiella* spp. isolated from chickens to be multidrug resistant, compared to 38.2% in our study. The discrepancies may also stem from the dissimilar periods of study (we noted significant differences over three consecutive years), the investigated poultry farming areas being distant from each other, and/or the production being managed on different systems [27,28,29].

It also cannot be ruled out that the changing proportions of strains resistant to different drugs in our study, especially in 2020 and 2021 (Figure 1), stem from reduced antibiotic use. As-yet-unpublished data [30] indicate that the amount of antibiotics per kg body weight used in turkey farming in the same area (north-eastern Poland) fell by 30% and 20% for male and female turkeys, respectively, from 2020 to 2021. This may partly explain the trends noted in the present study.

## 5. Conclusions

Multidrug-resistant strains of *Klebsiella* spp. pose a serious problem and risk to human health and life worldwide. Though our study suggests that the susceptibility to popular antimicrobials of analyzed *Klebsiella* spp. isolates identified between 2019 and 2021 has increased year by year, it is nevertheless necessary to continuously implement and enforce measures to minimize the evolution of multidrug-resistant bacteria and the build-up of antibiotic resistance, such as the rational use of antibiotics, targeted therapies, antibiotic-free rearing, improvement of zoohygienic conditions, and biosecurity compliance.

## Figures and Tables

**Figure 1 animals-12-03157-f001:**
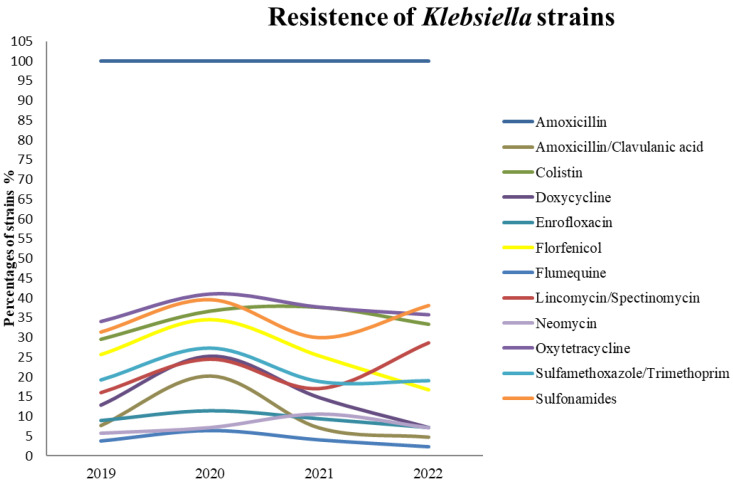
Percentage of antibiotic resistant *Klebsiella* strains in the consecutive years between 2019–2022.

**Table 1 animals-12-03157-t001:** Antibiotic susceptibility of *Klebsiella* spp. strains (n = 507) isolated from turkeys between 2019 and 2022.

Antibiotic	n	R	R%	I	I%	S	S%
Amoxicillin	507	507	100.00	0	0.00	0	0
Amoxicillin/Clavulanic acid	507	54	10.65	34	6.71	419	82.64
Colistin	507	23	4.54	13	2.56	471	92.90
Doxycycline	507	175	34.52	13	2.56	319	62.92
Enrofloxacin	507	83	16.37	104	20.51	320	63.12
Florfenicol	507	49	9.66	9	1.78	449	88.56
Flumequine	507	138	27.22	104	20.51	265	52.27
Lincomycin/Spectinomycin	507	100	19.72	22	4.34	385	75.94
Neomycin	507	40	7.89	10	1.97	457	90.14
Oxytetracycline	507	189	37.28	3	0.59	315	62.13
Sulfamethoxazole/Trimethoprim	507	108	21.30	3	0.59	396	78.11
Sulfonamides	507	171	33.73	5	0.99	331	65.29

n—number of samples; R-resistant strains; R%—percentage of resistant strains; I—intermediate strains; I%—percentage of intermediate strains; S—susceptible strains; S%—percentage of susceptible strains.

**Table 2 animals-12-03157-t002:** Antibiotic susceptibility of *Klebsiella* spp. strains isolated from turkeys in Poland.

Antibiotic	2019 (n = 156)						2020 (n = 139)							2021 (n = 170)					2022 (n = 42)					
	S	%	I	%	R	%	S	%	I	%	R	%	S	%	I	%	R	%	S	%	I	%	R	%
AMX ^1^	0	0.00	0	0.00	156	100.00	0	0.00	0	0.00	139	100.00	0	0.00	0	0.00	170	100.00	0	0.00	0	0.00	42	100.00
AMC	136	87.18	8	5.13	12	7.69	99	71.22	12	8.63	28	20.14	148	87.06	10	5.88	12	7.06	36	85.71	4	9.52	2	4.76
COL	148	94.87	2	1.28	6	3.85	122	87.77	8	5.76	9	6.47	160	94.12	3	1.76	7	4.12	41	97.62	0	0.00	1	2.38
DOX	104	66.67	6	3.85	46	29.49	84	60.43	4	2.88	51	36.69	104	61.18	2	1.18	64	37.65	27	64.29	1	2.38	14	33.33
ENR	103	66.03	33	21.15	20	12.82	79	56.83	25	17.99	35	25.18	107	62.94	38	22.35	25	14.71	31	73.81	8	19.05	3	7.14
FLO	131	83.97	1	0.64	14	8.97	118	84.89	5	3.60	16	11.51	151	88.82	3	1.76	16	9.41	39	92.86	0	0.00	3	7.14
UB	86	55.13	30	19.23	40	25.64	64	46.04	27	19.42	48	34.53	86	50.59	41	24.12	43	25.29	29	69.05	6	14.29	7	16.67
L/SPE	123	78.85	8	5.13	25	16.03	102	73.38	3	2.16	34	24.46	133	78.24	8	4.71	29	17.06	27	64.29	3	7.14	12	28.57
NEO	143	91.67	4	2.56	9	5.77	127	91.37	2	1.44	10	7.19	148	87.06	4	2.35	18	10.59	39	92.86	0	0.00	3	7.14
OT	103	66.03	0	0.00	53	33.97	80	57.55	2	1.44	57	41.01	105	61.76	1	0.59	64	37.65	27	64.29	0	0.00	15	35.71
SXT	126	80.77	0	0.00	30	19.23	100	71.94	1	0.72	38	27.34	136	80.00	2	1.18	32	18.82	34	80.95	0	0.00	8	19.05
SSS	100	64.10	0	0.00	49	31.41	81	58.27	3	2.16	55	39.57	117	68.82	2	1.18	51	30.00	26	61.90	0	0.00	16	38.10

^1^ AMX—Amoxicillin; AMC—Amoxicillin/Clavulanic acid; DOX—Doxycycline; ENR—Enrofloxacin; FLO—Florfenicol; UB—Flumequine; COL—colistine; L/SPE—Lincomycin/Spectinomycin; NEO—Neomycin; OT—Oxytetracycline; SXT—Trimethoprim/Sulfamethoxazole; SSS—Sulfonamides; S—sensitive; I—Intermediate; R—Resistant.

**Table 3 animals-12-03157-t003:** Evaluation of statistical significance between the number of *Klebsiella* spp. strains resistant to the tested antibiotics between 2019 and 2022. *p* < 0.05 indicates pairs of antibiotics that show statistically significant difference in the resistance rate.

Antibiotic	Amoxicillin n = 507	Amoxicillin/ Clavulanic acid n = 54	Colistin n = 23	Doxycycline n = 175	Enrofloxacin n = 83	Florfenicol n = 49	Flumequine n = 138	Lincomycin/ Spectinomycin n = 100	Neomycin n = 40	Oxytetracycline n = 189	Sulfamethoxazole/ Trimethoprim n = 108	Sulfonamides n = 171
Amoxicillin n = 507		*p* < 0.001	*p* < 0.001	*p* < 0.001	*p* < 0.001	*p* < 0.001	*p* < 0.001	*p* < 0.001	*p* < 0.001	*p* < 0.001	*p* < 0.001	*p* < 0.001
Amoxicillin/ Clavulanic acid n = 54			*p* = 0.0002	*p* < 0.001	*p* = 0.007	*p* = 0.6032	*p* < 0.001	*p* = 0.0001	*p* = 0.1295	*p* < 0.001	*p* < 0.001	*p* < 0.001
Colistin n = 23				*p* < 0.001	*p* < 0.001	*p* = 0.0015	*p* < 0.001	*p* < 0.001	*p* = 0.0270	*p* < 0.001	*p* < 0.001	*p* < 0.001
Doxycycline n = 175					*p* < 0.001	*p* < 0.001	*p* < 0.001	*p* < 0.001	*p* < 0.001	*p* = 0.3594	*p* < 0.001	*p* = 0.7911
Enrofloxacin n = 83						*p* = 0.0015	*p* < 0.001	*p* = 0.1651	*p* < 0.001	*p* < 0.001	*p* = 0.0447	*p* < 0.001
Florfenicol n = 49							*p* < 0.001	*p* < 0.001	*p* = 0.3179	*p* < 0.001	*p* < 0.001	*p* < 0.001
Flumequine n = 138								*p* = 0.0049	*p* < 0.001	*p* = 0.0006	*p* = 0.0280	*p* = 0.0244
Lincomycin/ Spectinomycin n = 100									*p* < 0.001	*p* < 0.001	*p* = 0.5338	*p* < 0.001
Neomycin n = 40										*p* < 0.001	*p* < 0.001	*p* < 0.001
Oxytetracycline n = 189											*p* < 0.001	*p* = 0.2375
Sulfamethoxazole/ Trimethoprim n = 108												*p* < 0.001
Sulfonamides n = 171												

## Data Availability

Data are contained within the article. The datasets used and/or analyzed during the current study are available from the corresponding author on reasonable request.

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
