# Peer review of "Identification and Antimicrobial Resistance in Klebsiella spp. Isolates from Turkeys in Poland between 2019 and 2022"

_animals, 2022, doi:10.3390/ani12223157_

Round 1

Reviewer 1 Report

The aim of this study was to show prevalence and antibiotic resistance of Klebsiella spp isolated from turkeys between 2019 and 2021.
The test methods used are correct, but Materials and Methods chapter requires additions. The discussion is well carried out and exhausting.
References well chosen. Before publishing in Animals, the paper requires additions and corrections.
The list of proposed changes is given below:
General comments:
1. Bacterial names must alweys be italicised
- correct in the entire text

2.     Suggestion to the authors to add detail for the sample size in Materials and Methods unit e.g. how many samples were from the post mortem how many from live birds?

3.     There is "prevalence" in the title so the results and discussion section should be supplemented with this data by year.

4.     Line 158-159: This sentence should be omitted

5.     Line 93: should be  37°C

6.     In Discussion unit – include references to the tables and figure.

So, after revision the paper has the potential to be published in this journal.

Author Response

1) English language and style are fine/minor spell check required

Response 1: Manuscript before submitting was proofread by a native speaker of English who has rendered proofreading services in the veterinary medicine field.

2) Bacterial names must alweys be italicised - correct in the entire text

Response 2: The authors would like to thank the Reviewer for this comment. The relevant correction has been made in the revised MS.

3) Suggestion to the authors to add detail for the sample size in Materials and Methods unit e.g. how many samples were from the post mortem how many from live birds?

Response 3: The authors would like to thank the Reviewer for this suggestion. There were 153 samples from live birds as palatine fissure swabs or trachea swabs and 354 samples from post-mortem (from the lungs, heart, liver, spleen, suborbital sinuses, yolk sac, and joints). The relevant correction has been made in the revised MS (line 90-91).

4) There is "prevalence" in the title so the results and discussion section should be supplemented with this data by year.

The authors thank the reviewer for this comment. The number of orders for microbiological analysis of samples from turkeys in the SLW Biolab laboratory is quite large, and at this point it is possible to give an estimate. For this reason, the authors propose to change the title of the manuscript from "Prevalence and antibiotic resistance...". to "Identification and antibiotic resistance...".  The title of the paper for rewritten in order to meet reviewer’s recommendation.

5) Line 158-159: This sentence should be omitted

Response 5: The authors would like to thank the Reviewer for this comment. The final sentence of the results has been deleted.

6) Line 93: should be 37°C

Response 6: The relevant correction has been made in the revised MS (line 93).

7) In Discussion unit – include references to the tables and figure.

Response 7: The authors would like to thank the Reviewer for this suggestion. The relevant correction has been made in the revised MS (line 90-91).

Reviewer 2 Report

This is an interesting article, which focuses on drug resistance within turkeys, which compared to chickens is a very understudies area. I enjoyed reading it and am pleased to see more studies on AMR in animals, particularly from an angle of Klebsiella which is rarely studied. So I commend the authors on their studies and the results, which are both interesting and scary.

I do however have a few comments which need to be addressed prior to acceptance, not least with the use of italics for bacterial names, and the lack of controls used.

There are many areas in the document where bacterial names are not correctly italicised. I have tried to list these here, but this is not an exhaustive list. Also in vitro and in vivo should be italicised.  

Lines 44, 47, 51, 52, 53, 130, 132, 133, 130, 138, 142, 177

Also, you have 100% resistance to amoxicillin, but you don’t use any controls to confirm that the antibiotic is actually working. Perhaps testing the antibiotic against an ATCC E. coli or Klebsiella strain may be useful to confirm that the antibiotic is working. I feel that you have probably already done this, just haven’t reported it.

You also talk in several places, particularly 144-160 about statistically significant differences, but do not include p values. Please can these be included.

Specific comments

Line 17- maybe worth mentioning veterinarians as well as physicians?

Line 30- antibiotic resistant bacteria ?

Line 37- you talk about a change for specific drug groups, maybe worth giving some examples in here  

Line 98- identification of Klebsiella species.

Lines 103-106- the antibiotic concentrations need to be mentioned here.

Line 112- delete ‘the active substances of’ ….

Line 125-126- Not completely sure that makes full sense- please reword

Line 131- this would read better as ‘95% of the strains were ….’

Line 146- is this significant difference related to the collection/ isolate numbers per year?

Figure 1- the graph doesn’t need a title

Line 196- effective against Klebsiella spp. ….. (delete ‘the’)

Line 198- nice comparison but maybe worth saying in which species and country this was.

Line 214- this seems a slightly odd statement. Do you have sex of the birds in this study? And if so, is there a sex difference? And why were male and female drug use levels different?

Line 220- when you say more susceptible, what do you mean? More susceptible than what? And how would that happen?

Author Response

1) Moderate English changes required

Response 1: Manuscript before submitting was proofread by a native speaker of English who has rendered proofreading services in the veterinary medicine field.

2) There are many areas in the document where bacterial names are not correctly italicised. I have tried to list these here, but this is not an exhaustive list. Also in vitro and in vivo should be italicised.  

Response 2: The authors thank the Reviewer for drawing attention to the editorial mistakes. The relevant corrections have been made in the revised MS.

3) Also, you have 100% resistance to amoxicillin, but you don’t use any controls to confirm that the antibiotic is actually working. Perhaps testing the antibiotic against an ATCC E. coli or Klebsiella strain may be useful to confirm that the antibiotic is working. I feel that you have probably already done this, just haven’t reported it.

Response 3: The authors would like to thank the Reviewer for this comment and for the order Enterobacterales the following reference strains were used: Escherichia coli ATCC 25922 (main quality control); Escherichia coli ATCC 35218 (for beta-lactamase inhibitor control, ESBL negative); Klebsiella pneumoniae subsp. pneumoniae ATCC 700603 (ESBL positive). The relevant correction has been made in the revised MS (line 122-125).

4) You also talk in several places, particularly 144-160 about statistically significant differences, but do not include p values. Please can these be included.

Response 4: The authors would like to thank the Reviewer for this comment, but all p-values are presented in Table 3 entitled: Evaluation of statistical significance between the number of Klebsiella spp. strains resistant to the tested antibiotics between 2019-2022. P<0.05 indicates pairs of antibiotics that show statistically significant difference in the resistance rate. The authors did not want to copy results in text and decided only to present them in table (which, in authors’ opinion, seems more transparent of relevant data for readers).

5) Line 17- maybe worth mentioning veterinarians as well as physicians?

Response 5: The authors would like to thank the Reviewer for this suggestion. The relevant correction has been made in the revised MS (line 17).

6) Line 30- antibiotic resistant bacteria ?

Response 6: The authors thank the Reviewer for drawing attention to the editorial mistake. The relevant correction has been made in MS.

7) Line 37- you talk about a change for specific drug groups, maybe worth giving some examples in here 

Response 7:

8) Line 98- identification of Klebsiella species.

Response 8: The authors thank the Reviewer for drawing attention to the editorial mistake. The relevant correction has been made in MS.

9) Lines 103-106- the antibiotic concentrations need to be mentioned here.

Response 9) The authors would like to thank the Reviewer for this comment and the relevant correction has been made in MS (line 104-109).

10) Line 112- delete ‘the active substances of’ ….

Response 10: The authors would like to thank the Reviewer for this suggestion. The relevant correction has been made in the revised MS.

11) Line 125-126- Not completely sure that makes full sense- please reword

12) Line 131- this would read better as ‘95% of the strains were ….’

Response 10: The authors would like to thank the Reviewer for this suggestion. The relevant correction has been made in the revised MS.

13) Line 146- is this significant difference related to the collection/ isolate numbers per year?

14) Figure 1- the graph doesn’t need a title

Response 14: The authors would like to thank the Reviewer for this comment but according to the Instructions for Authors “All Figures, Schemes and Tables should have a short explanatory title and caption.”

15) Line 196- effective against Klebsiella spp. ….. (delete ‘the’)

Response 15: The authors would like to thank the Reviewer for this suggestion. The relevant correction has been made in the revised MS. (line 201)

16) Line 198- nice comparison but maybe worth saying in which species and country this was.

17) Line 214- this seems a slightly odd statement. Do you have sex of the birds in this study? And if so, is there a sex difference? And why were male and female drug use levels different?

18) Line 220- when you say more susceptible, what do you mean? More susceptible than what? And how would that happen?
